# Genome-Wide Analysis Identifies Candidate Genes Encoding Beak Color of Duck

**DOI:** 10.3390/genes13071271

**Published:** 2022-07-18

**Authors:** Qixin Guo, Yong Jiang, Zhixiu Wang, Yulin Bi, Guohong Chen, Hao Bai, Guobin Chang

**Affiliations:** 1Joint International Research Laboratory of Agriculture and Agri-Product Safety, The Ministry of Education of China, Yangzhou University, Yangzhou 225009, China; dx120190114@yzu.edu.cn (Q.G.); jiangyong@yzu.edu.cn (Y.J.); wangzx@yzu.edu.cn (Z.W.); ylbi@yzu.edu.cn (Y.B.); ghchen2019@yzu.edu.cn (G.C.); 2College of Animal Science and Technology, Yangzhou University, Yangzhou 225009, China

**Keywords:** duck, beak color, GWAS, melanin

## Abstract

Beak color diversity is a broadly occurring phenomenon in birds. Here, we used ducks to identify candidate genes for yellow, black, and spotted beaks. For this, an F_2_ population consisting of 275 ducks was genotyped using whole genome resequencing containing 12.6 M single-nucleotide polymorphisms (SNPs) and three beak colors. Genome-wide association studies (GWAS) was used to identify the candidate and potential SNPs for three beak colors in ducks (yellow, spotted, and black). The results showed that 2753 significant SNPs were associated with black beaks, 7462 with yellow, and 17 potential SNPs with spotted beaks. Based on SNP annotation, *MITF*, *EDNRB2*, members of the POU family, and the SLC superfamily were the candidate genes regulating pigmentation. Meanwhile, isoforms MITF-M and *EDNRB2* were significantly different between black and yellow beaks. MITF and EDNRB2 likely play a synergistic role in the regulation of melanin synthesis, and their mutations contribute to phenotypic differences in beak melanin deposition among individuals. This study provides new insights into genetic factors that may influence the diversity of beak color.

## 1. Introduction

Beak pigmentation is a common phenomenon observed in most birds. The color of feathers, coats, and skin is primarily determined by melanocytes, which are involved in the synthesis of melanin and play an important role in cosmetic change, heat regulation, camouflage, and protection against UV radiation from sun exposure [1,2,3]. In addition, melanin accumulation can lead to hyperpigmentation of the skin [4,5]. Previous studies have shown that skin color is highly heritable and one of the most variable phenotypes. This phenotype is influenced not only by genetic factors, but also by the environment [6,7]. Skin pigmentation is highly associated with latitude and fundamentally, the distribution of ultraviolet (UV) radiation [8]. Skin pigmentation is also influenced by the concerted action of different types of natural selection, including climate, lifestyle, diet, and metabolism [9,10].

With rapid development in genetics and genomics, researchers have gradually realized that human skin color diversity is due to the natural positive selection of those genes that affect human pigmentation, especially in melanosome biogenesis or melanin biosynthetic pathways [11,12,13,14]. Recently, a large number of genome-wide association studies (GWAS) of pigmentation have established that some single-nucleotide polymorphisms (SNPs) in *TYR*, *IRF4*, *TYRP1*, *OCA2*, *SLC45A2*, *MC1R*, and *KITLG* genes are significantly associated with human skin color [15,16,17]. Moreover, the α-MSH gene is a significant inherited factor that acts mainly as an agonist of *MC1R* [18]. Furthermore, SLC45A2 (also known as *AIM1* or *MATP*) encodes a transporter that mediates melanin synthesis and is expressed in a high percentage of melanoma cell lines [19,20]. Several *SLC45A2* mutations have been reported to lead to *OCA4,* and polymorphisms of this gene are significantly associated with human skin, hair, and eye pigmentation [21,22,23]. Selected signatures of skin pigmentation loci have been revealed by studies of modern and ancient populations, with some genes showing variation associated with light skin pigmentation also showing polygenic selection in western Eurasia [24]. However, this observation is the only recorded sign of polygenic selection for skin pigmentation based on only four loci (*SLC24A5*, *SLC45A2*, *TYR*, and *APBA2*/*OCA2*) [25]. 

Ducks, the second-largest poultry species in the world, mostly have yellow and black beaks. Occasionally, spotting occurs, which directly affects carcass sales. However, genetic factors that lead to the appearance of spot color remain unclear. Based on an F_2_ cross between the Cherry Valley Duck and Runzhou White Crested Duck, we performed a genome-wide association study (GWAS) with 275 birds to gain insight into the effects of genetic factors on beak pigmentation. These studies provided insight into the molecular regulatory mechanisms and genetic improvement of melanin deposition in duck beaks. 

## 2. Materials and Methods

### 2.1. Ethical Approval

All experiments on ducks were performed in accordance with the Regulations on the Administration of Experimental Animals issued by the Ministry of Science and Technology (Beijing, China) in 1988 (last modified in 2001). The experimental protocols were approved by the Animal Care and Use Committee of the Yangzhou University (YZUDWSY2017-11-07). All efforts were made to minimize animal discomfort and suffering.

### 2.2. Samples and Sequencing

The F_2_ resource population, which crosses the Chinese Crested duck (CC duck) and Cherry Valley duck (CV duck), was obtained from the Laboratory of Poultry Genetic Resources Evaluation and Germplasm Utilization at Yangzhou University. The ducks were raised in stair-step cages under the recommended environmental and nutritional conditions at the conservation farm of Ecolovo Group, China. The CC duck is a Chinese indigenous breed that has a black shank and beak, while the CV duck is a commercial breed that has a yellow shank and beak and white plumage. In the F_1_ generation, 30 CC ducks and six CV ducks were randomly selected and divided into six families to interbreed. To generate F_2_ progeny, 30 males and 150 unrelated female ducks were used as hybrids. A total of 275 ducks were used in the next experiment. To identify candidate genes associated with beak color, we classified beak color as yellow, spotted, or black. The R/ggcor package was used to calculate the correlation between beak color and sex.

Blood samples were used to collect high-quality DNA at 42 days of age. Genomic DNA (gDNA) was extracted from blood samples by using the DNA extraction kit method (QIAampR DNA Blood Mini Kit), following the manufacturer’s protocol. Two paired-end sequencing libraries with insert sizes of 350 bp were constructed according to the Illumina protocol (Illumina, San Diego, CA, USA). All libraries were sequenced using the Illumina NovaSeq platform.

### 2.3. Genotyping

Raw reads were filtered using the NGS QC Toolkit (version 2.3) with default parameters [26]. The clean reads were mapped to the duck reference genome (the Chinese Crested duck genome was assembled by our lab (unpublished) with a Burrows–Wheeler alignment (BWA aln) using the default parameters) [27]. GATK then performed SNP calling [28]. VCF tools were used to further filter the raw data [29]. The SNPs were screened with parameters of a minor allele frequency (MAF) > 0.05, maximum allele frequency < 0.99, and maximum missing rate < 0.01. After filtering, SNPs showed a mean density of 8.5 SNPs/kb across the genome. All filtered SNPs were distributed on 37 autosomal chromosomes, ChrZ, and ChrU (unplaced scaffolds). 

### 2.4. Population Structure

The population structure was assessed with multidimensional scaling (MDS) using PLINK 1.9 software [30]. Independent SNPs were obtained on all autosomes using the in-dep-pairwise option, with a 1000 bp window, five steps, and an r2 threshold of 0.2. Pairwise identity-by-state (IBS) distances between all individuals were calculated using these independent SNP markers, and MDS components were acquired using the mds-plot option based on the IBS matrix. A relative kinship matrix was constructed using these independent SNP markers.

### 2.5. Whole-Genome Association Analysis and Linkage Disequilibrium Analysis

The GWAS analysis of beak color used the linear mixed model in the Effective Mixed Model Association eXpedited (EMMAX) software [31]. EMMAX makes the simplifying and time-saving assumption that any given SNP’s effect on the trait is typically small, and therefore only estimates the model variance components once per analysis to account for population structure. EMMAX estimates variance components using the REML model.
y=Xa+Zb+e
where *y* is a vector of beak colors, *X* is the incidence matrix for a random additive effect, *a* is the column vector of random additive effects, *Z* is the genotype value of the candidate SNP, *b* is the regression coefficient of the candidate SNP, and *e* is the random residual. The phenotypic variance–covariance matrix is *var*(*y*) = *Var*(*a*) + *var*(*e*) = *K σ_a_^2^* + *I σ_e_^2^*, where *K* is the IBS kinship matrix, *I* is the identity matrix, *σ_a_^2^* is the additive variance, and *σ_e_^2^* is the variance of random residuals. The regional Manhattan plot and LD heatmap were obtained using LDBlockShow software. 

### 2.6. Gene Ontology (GO) and Kyoto Encyclopedia of Genes and Genomes (KEGG) Analyses

Based on the LD attenuation distance calculated using PopLDdecay [32], annotation of related genes in a certain region upstream and downstream of the physical location of the significant SNPs were performed. The sequences of the relevant genes were extracted from the mallard genome and translated into a protein sequence, which was then analyzed using KOBAS 3.0 software [33]. Chicken was selected as the reference species, and hypergeometric tests along with Fisher’s exact test were used as the statistical methods.

## 3. Results

### 3.1. Phenotypic Description and Population Structure Analysis

To identify the candidate genes for beak color, we first focused on the correlation between beak color and sex to determine whether there was a correlation between these two traits. The results showed that there was no correlation between black, yellow, or spotted beak color and sex (Figure 1a). We also identified the population structure of all the samples used in the present study using MDS. The results showed that the three different beak-colored ducks had no obvious clustering and were evenly distributed (Figure 1b).

### 3.2. Genome-Wide Association Study Identified the Candidate Variants for Beak Color

EMMAX software was used to conduct the genome-wide association analysis in the present study. The Q–Q plot illustrated that the model used for the GWAS analysis was reasonable. The lambdas (inflation factor (λ)) of the three different color beaks were 0.98 (black beak), 1.05 (spotted beak), and 0.99 (yellow beak), and the points at the upper right corner of the Q–Q plots were the significant markers associated with the traits under study (Figure 2). Thus, population stratification was adequately controlled. However, no significantly associated sites were found in the GWAS analysis of spotted beaks in the Q–Q plot, but we found a large number of potential associated sites for spotted beak through the Manhattan plot. 

The Manhattan plot of beak color showed that a total of 2753 significant SNPs associated with black beak were identified using the threshold of significant *p*-value (threshold = 0.05/total number of all SNPs = 3.94885 × 10^−09^), and 1916 extremely significant SNPs were identified using the threshold of significant *p*-value (threshold = 0.01/total number of all SNPs = 7.8977 × 10^−10^), most of which were located on chromosome 14 (ALP 14) (2708 SNPs) and ALP 11 (45 SNP) (Figure 3, top). 

For yellow beak, 7462 significant SNPs were identified using the threshold of significant *p*-value (threshold = 0.05/total number of all SNPs = 3.94885 × 10^−09^), and 5878 extremely significant SNPs were identified using the threshold of significant *p*-value (threshold = 0.01/total number of all SNPs = 7.8977 × 10^−10^), most of which were located in ALP11 (Figure 3, middle). 

For spotted beaks, there were no significant SNPs associated with them (*p*-value ≤ 3.94885 × 10^−09^). However, we identified 17 potential SNPs associated with spotted beaks using the threshold of significant *p*-value (threshold = 1/total number of all SNPs = 7.8977 × 10^−08^) (Figure 3, bottom). We found 45 shared SNPs between black and yellow beaks, 11 shared SNPs between black and spotted beaks, and 5 shared SNPs between yellow and spotted beaks using a Venn analysis (Figure 4). In addition, based on the result of all SNP synonymous analysis that all exonic SNPs were synonymous and not predicted to alter protein function.

### 3.3. Functional Analysis 

The genes or genomic regions identified in the GWAS explained only part of the genetic variation. To overcome this limitation, the GWAS was complemented with a gene set analysis using GO and KEGG databases to detect potential functional categories underlying the beak color. Based on the SNP annotation, we found that 94 genes, including *MITF*, *EDNRB2*, *SLC25A43*, *SLC25A5*, *SLC25A14*, *SPRY3*, *POU4F3*, etc. (Appendix A), were the most significant candidates associated with a black beak. The results of the KEGG and GO enrichment analysis showed that these candidate genes were involved in melanogenesis, necroptosis, calcium signaling, the FoxO signaling pathway, primary bile acid biosynthesis, apoptosis, the positive regulation of secretion by cells, positive regulation of secretion, and the mitochondrial part (Figure 5a). In addition, *MITF*, *POU4F1*, *POU3F3B*, *POU1F1*, *POU2F1*, *SLC7A1*, *SLC46A3*, *SLC25A15*, *SLC25A30*, *SLC15A1*, *SLC10A2*, *SLC5A7*, *SLC9A2*, *SLC9A4,* and others were most significantly associated with yellow beak (Appendix A). Cytokine–cytokine receptor interaction, apoptosis, phototransduction, the thyroid hormone signaling pathway, DNA replication, axon guidance, the cell adhesion molecule pathway, cytokine receptor activity, lipid-transporting ATPase activity, gap junction channel activity, negative regulation of adherens junction organization, wide pore channel activity, axon choice point recognition, axon midline choice point recognition, negative regulation of negative chemotaxis, cellular response to vitamin K, positive regulation of small-molecule metabolic processes, microtubule cytoskeleton, and pigmentation were enriched (Figure 5b). Finally, we found *MITF*, *TMLHE*, *EDNRB2*, *NUP98*, *ITPR1*, *CHL1*, *ALG8*, *SPRY3*, and *PDHB* genes (Appendix A), which are involved in melanogenesis, the calcium signaling pathway, the citrate cycle (TCA cycle), lysine degradation, etc.; and dolichyl pyrophosphate Glc1Man9GlcNAc2 α-1,3-glucosyltransferase activity, trimethyllysine dioxygenase activity, inositol 1,4,5-trisphosphate receptor activity involved in regulation of postsynaptic cytosolic calcium levels, pyruvate dehydrogenase (acetyl-transferring) activity, inositol 1,4,5-trisphosphate-sensitive calcium-release channel activity, the carnitine biosynthetic process, α-1,3-mannosyltransferase activity, endothelin receptor activity, the amino acid betaine biosynthetic process, and regulation of postsynaptic cytosolic calcium ion concentration terms were the potential candidate genes for spotted beak (Figure 5c).

### 3.4. The EDNRB2 and MITF Isoform Expression Level in Black and Yellow Beaks

The genes responsible for melanoblast migration and melanocyte development include *EDN3*, *EDNRB*, *EDNRB2*, *MITF*, *KIT*, and *KITLG* [34]. Based on the GWAS analysis, we found that *MITF* and *EDNRB2* were candidate genes for beak color, and therefore might regulate pigmentation. Duck *MITF* consists of two isoforms, MITF-B and MITF-M, with isoform-specific first exons called 1B and 1M, respectively (Figure 6a). Thus, we determined the expression levels of these two isoforms in black and yellow beaks. The results showed that the MITF-M was significantly expressed in black beaks (Figure 6b). The *EDNRB2* expression levels were compared between black and yellow beaks. *EDNRB2* expression levels showed that *EDNRB2* was significantly expressed in black beaks (Figure 6b).

## 4. Discussion

Melanin is a substance in the body that is responsible for hair, eye, and skin pigmentation [35]. Melanin is a complex polymer that originates from the amino acid tyrosine [36]. Melanin is present in human and animal skin to varying degrees, and is responsible for unique eye, hair, and skin colors [4,35,37]. The color of a bird’s beak, which is the exposed skin tissue, results from concentrations of pigments, primarily melanin and carotenoids, in the epidermal layers, including the rhamphotheca [38]. In duck beaks, melanin deposition increases with age and UV exposure [39,40]. However, under the influence of domestication and selection, many duck breeds have a significantly fixed and stable inheritance of the black beak, as seen in Chinese Crested ducks and Lianchen ducks. Most duck breeds exhibit yellow beaks. However, some duck breeds also exhibit spotted beaks. To determine the genetic basis of beak color diversity, the present study designed an F_2_ population cross between Chinese Crested ducks and Cherry Valley ducks. A GWAS was performed to identify candidate genes associated with different beak colors. In black beaks, we found two significantly associated signals: *MITF* and *EDNRB2*. Most genes belonging to the SLC superfamily and POU (Pit-Oct-Unc) family were significantly associated with yellow beaks. Although we did not identify a significantly associated signal in the spotted beak, we found two candidate signal loci on chromosomes 11 and 14. By annotating the candidate signals, we found that *MITF* and *EDNRB2*, two key genes responsible for melanin synthesis, were enriched. 

*MITF* has been shown to affect pigmentation in cattle [41,42,43,44], mice [45,46,47], horses [48,49,50,51], dogs [52,53], humans [54,55], and ducks [56]. *MITF* belongs to the basic helix–loop–helix–leucine zipper (bHLHZip) family of proteins. Studies have shown that it regulates melanogenesis by binding to the highly conserved M-box (GTCATGTGCT) and E-box (TCATGTG) motifs upstream of the tyrosinase promoter, thereby strongly stimulating and promoting the activity of the tyrosinase promoter. Tyrosinase expression promotes melanin production [57,58,59,60]. Our results implied that melanin synthesis and metabolic pathways play crucial roles in inducing melanin deposition in beaks and genes related to melanin synthesis and metabolic pathways, such as *MITF*, *MC1R*, *EDNRB2*, the PMEL family, *TYR*, *TYRP1*, and *TYRP2*, affect melanin syntheses. Our findings showed similar results, including a significant association of *MITF*. 

*EDNRB* is a gene expressed in melanocytes that are derived from the neural crest, and for this reason, EDNRB is particularly mentioned. *EDNRB2* is a homolog of EDNRB, which belongs to the endothelin receptor (EDNR) gene family and has been lost in mammalian lineages. EDNRB signaling is required for melanocyte development [61,62]. Loss of function variants in EDNRB leads to white spotting phenotypes in humans, animals [63,64], and poultry [65,66]. A similar result was identified in our study, which showed that *EDNRB2* regulated melanin pigmentation in ducks. In addition, our results showed that the spotted beak was mainly coregulated by *MITF* and *EDNRB2*. However, the mechanism by which *MITF* and *EDNRB2* are coregulated requires further investigation.

Our results suggested that the POU family and SLC superfamily are significantly associated with yellow beaks. Yellow skin, beaks, and feet in most birds are caused by carotene deposits. The results showed that most members of the POU family, which share the typical POU domain structure [67], were significantly associated with yellow beaks. The POU family is a transcription factor family member that can promote the transcription of many genes related to development and metabolism, especially in Schwann and progenitor cell development [68,69]. Thirteen POU gene family members were found in the duck genome. In the present study, 44 POU transcription factors were predicted to be distributed within the promoter region of *MITF* [40], and *POU4F1*, *POU3F3B*, *POU1F1*, and *POU2F1* were identified. Therefore, we speculated that the yellow beak is coregulated by the POU family and *MITF*, but the specific mechanism requires further experimental verification. This study provided new clues for understanding the genetic factors that may influence the diversity of beak color, but further experimental studies are needed to strengthen this hypothesis.

## 5. Conclusions

This study identified candidate genes closely related to duck beak color. *MITF* and *EDNRB2* were the candidate genes associated with beak melanosis. We speculated that beak pigmentation may be coregulated by the POU family, *MITF,* and *EDNRB2*. However, the specific mechanisms require further experimental verification.

## Figures and Tables

**Figure 1 genes-13-01271-f001:**
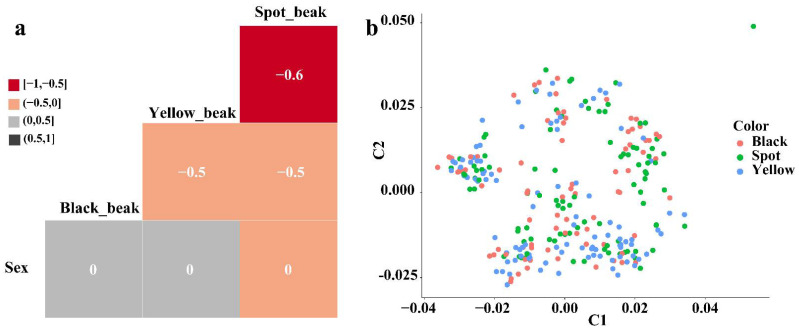
Beak color and sex correlation analysis (**a**); population structure analysis (**b**).

**Figure 2 genes-13-01271-f002:**
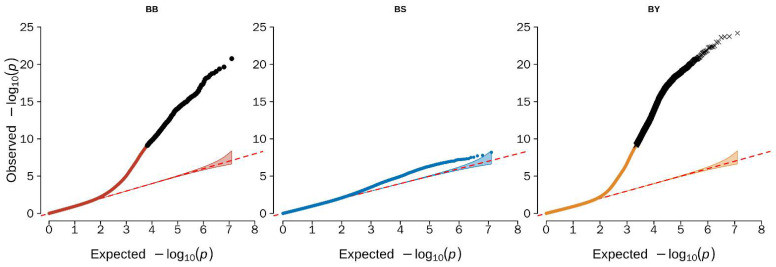
Quantile–quantile (Q–Q) from GWAS for beak color trait in duck. Q–Q plot showing the late separation between observed and expected values. The red lines indicate the null hypothesis of no true association. Deviation from the expected *p*-value distribution is evident only in the tail area for each trait, indicating that population stratification was properly controlled. BB refers to black beak; BS refers to spotted beak; BY refers to yellow beak.

**Figure 3 genes-13-01271-f003:**
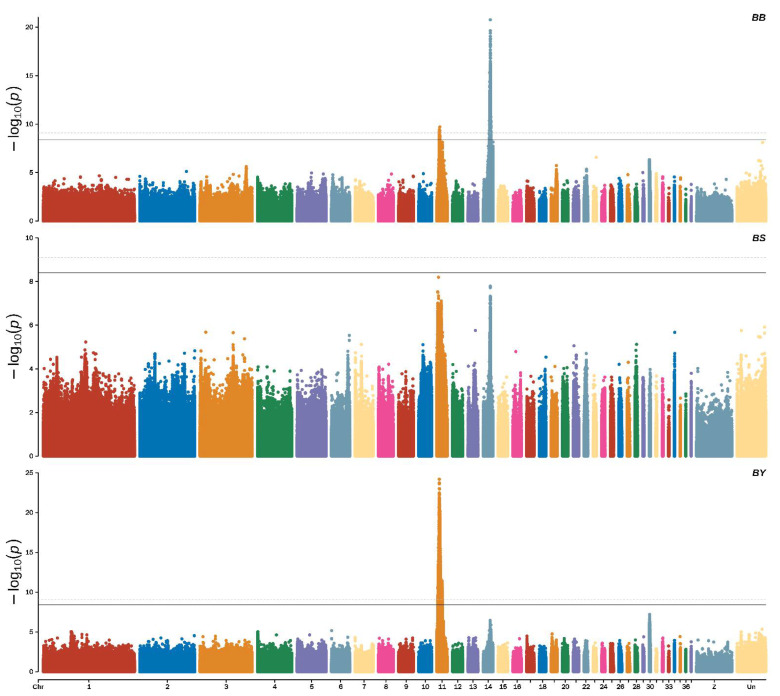
Manhattan plots showing the significance of genetic effects on the beak color according to the GWAS.

**Figure 4 genes-13-01271-f004:**
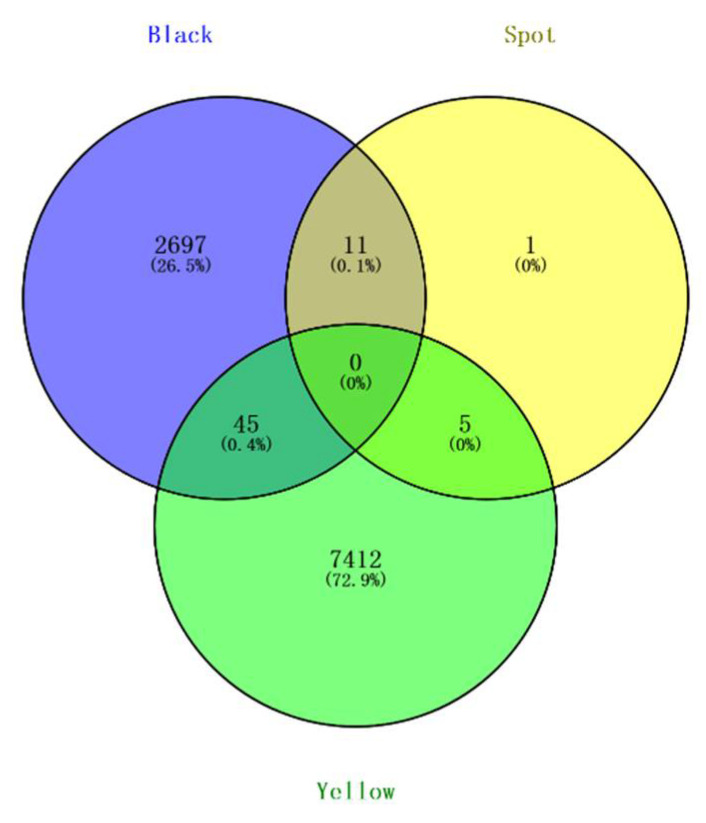
Venn analysis of all beak colors showing overlap of significant SNPs.

**Figure 5 genes-13-01271-f005:**
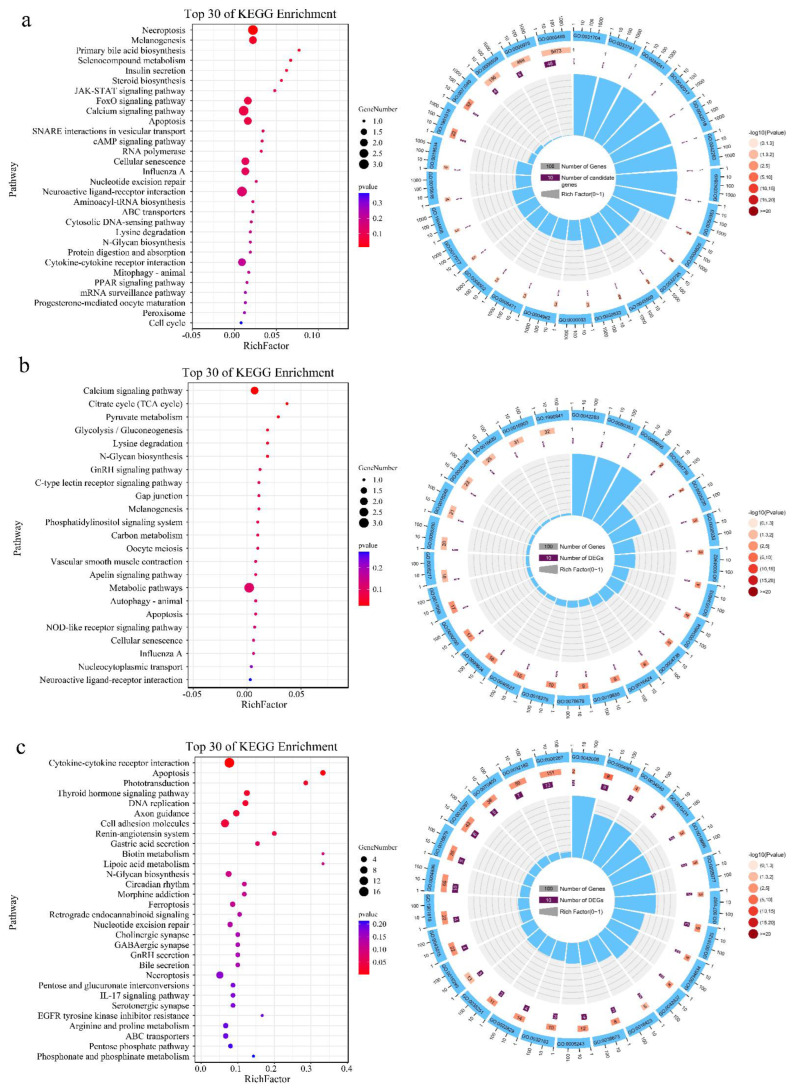
Functional enrichment analysis of the beak color candidate genes. (**a**) KEGG (left) and GO (right) enrichment of black beak candidate genes; (**b**) KEGG (left) and GO (right) enrichment of spotted beak candidate genes; (**c**) KEGG (left) and GO (right) enrichment of yellow beak candidate genes.

**Figure 6 genes-13-01271-f006:**
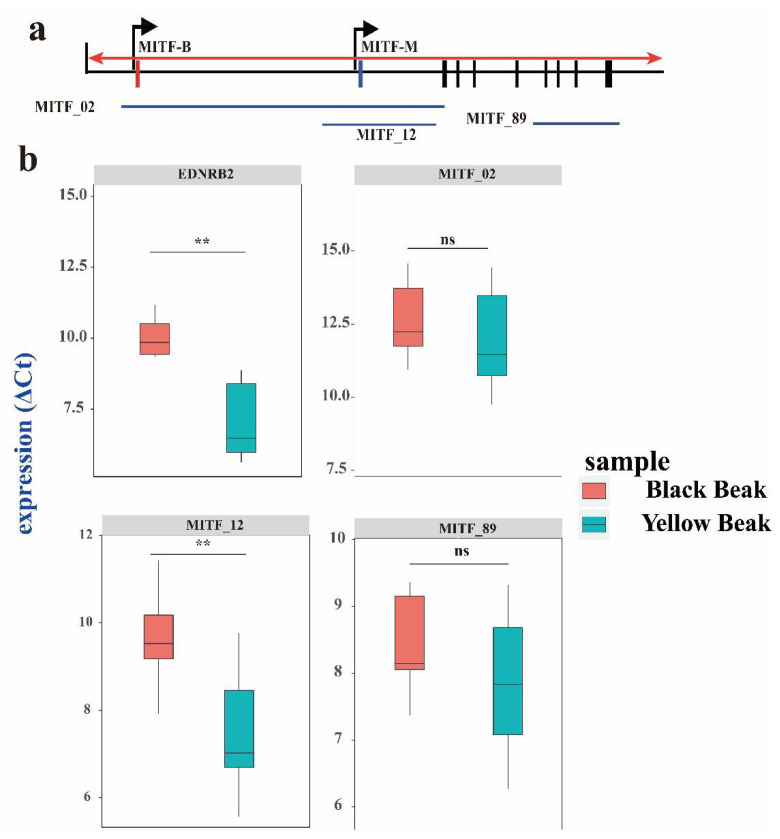
Expression differences in *EDNRB2* and *MITF* on three exon junctions between black and yellow beaks according to RT-qPCR. (**a**) Information on the *MITF* isoform. The red triangle represents the intronic insertion on chromosome 13 in Pekin ducks. Exon 1M is specific for the MITF-M transcript, while exon 1B is specific for the MITF-B transcript. (**b**) *EDNRB2* and *MITF* on three exon junctions between black and yellow beaks. Each exon junction was assayed in six biological replicates with three technical replicates. The indicated *p*-values were based on one-way ANOVA. NS, nonsignificant; **, extremely significant.

## Data Availability

The genome assembly and all of the resequencing data used in this research were deposited in the Genome Sequence Archive (GSA) at National Genomics Data Center (http://bigd.big.ac.cn (accessed on: 5 February 2022)) Beijing Institute of Genomics, Chinese Academy of Sciences (GSA: CRA005019).

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
