# Peer review of "Genome-Wide Analysis Identifies Candidate Genes Encoding Beak Color of Duck"

_genes, 2022, doi:10.3390/genes13071271_

Round 1

Reviewer 1 Report

Generally, the content of the presented publication does not raise any major reservations. Nevertheless, the authors should explain, for example, in the introduction why the test samples came from the collected blood and not, for example, from goose skin sections of different colors or from feather samples?

Author Response

Generally, the content of the presented publication does not raise any major reservations. Nevertheless, the authors should explain, for example, in the introduction why the test samples came from the collected blood and not, for example, from goose skin sections of different colors or from feather samples?

Response: Thank you for your acknowledgement of my article and for your comments. I am not sure I understood your comment correctly. The main aim of the current study is to reveal potential candidate genes for different beak colors through the GWAS approach. Beak color may be equally influenced by environmental factors, such as UV light, etc. These environmental factors may lead to some mutations in the DNA of skin cells. Therefore, the best sample used to collect DNA is blood. Thanks!

Reviewer 2 Report

The presented work “Genome-wide analysis identifies candidate genes encoded beak color of duck “deals with the colour of the beak of duck. The authors performed a genome-wide association study (GWAS) to gain insight into the effects of genetic factors on beak pigmentation. These studies provide insight into the molecular regulatory mechanisms and genetic improvement of melanin deposition in duck beaks.

The work is written at a high professional level.

I have the following questions or suggestions:

·       The whole work is devoted to the study of beak coloration, but in line 106 write: GWAS analysis for plumage color used the linear mixed model by the Effective Mixed Model Association eXpedited (EMMAX) software.

·       Similarly in a rows 137,152 instead of plumage to use beak

·       I recommend extending the conclusions of the work

Author Response

The presented work “Genome-wide analysis identifies candidate genes encoded beak color of duck “deals with the colour of the beak of duck. The authors performed a genome-wide association study (GWAS) to gain insight into the effects of genetic factors on beak pigmentation. These studies provide insight into the molecular regulatory mechanisms and genetic improvement of melanin deposition in duck beaks.

 The work is written at a high professional level.

I have the following questions or suggestions:

  • The whole work is devoted to the study of beak coloration, but in line 106 write: GWAS analysis for plumage color used the linear mixed model by the Effective Mixed Model Association eXpedited (EMMAX) software.
  • Similarly in a rows 137,152 instead of plumage to use beak
  • I recommend extending the conclusions of the work

Response: Thank you very much for your acknowledgement and very careful examination of my manuscript. I have made changes to the errors in the manuscript.